# T-Cell Engagers—The Structure and Functional Principle and Application in Hematological Malignancies

**DOI:** 10.3390/cancers16081580

**Published:** 2024-04-20

**Authors:** Paweł Cech, Katarzyna Skórka, Laura Dziki, Krzysztof Giannopoulos

**Affiliations:** Department of Experimental Hematooncology, Medical University of Lublin, 20-093 Lublin, Poland; cech.pawel@gmail.com (P.C.); laura52757@gmail.com (L.D.); krzysztof.giannopoulos@umlub.pl (K.G.)

**Keywords:** T-cell engager (TCE), bi-specific antibody (BsAb), bi-specific T-cell engager (BiTE), blinatumomab, T-cell engaging therapies, immunotherapies in hematology

## Abstract

**Simple Summary:**

Recent advancements in cancer research have proven immunotherapies to be a promising strategy for the treatment of hematological malignancies. The bispecific antibody (BsAb) format was developed to overcome the issues of monoclonal antibody-based therapies. T-cell engagers (TCEs) are BsAbs, which directly activate T-cells and their anti-tumor features, ultimately resulting in the lysis of the targeted tumor cells. In 2014, the FDA approved blinatumomab for treatment of acute lymphoblastic leukemia. As of November 2023, seven clinically approved TCE therapies are on the market. In this paper, we summarized the technical basis of the TCE technology, its application in hematology, and its current issues and prospects.

**Abstract:**

Recent advancements in cancer immunotherapy have made directing the cellular immune response onto cancer cells a promising strategy for the treatment of hematological malignancies. The introduction of monoclonal antibody-based (mAbs) targeted therapy has significantly improved the prognosis for hematological patients. Facing the issues of mAb-based therapies, a novel bispecific antibody (BsAb) format was developed. T-cell engagers (TCEs) are BsAbs, which simultaneously target tumor-associated antigens on tumor cells and CD3 molecules present on T-cells. This mechanism allows for the direct activation of T-cells and their anti-tumor features, ultimately resulting in the lysis of tumor cells. In 2014, the FDA approved blinatumomab, a TCE directed to CD3 and CD19 for treatment of acute lymphoblastic leukemia. Since then, numerous TCEs have been developed, allowing for treating different hematological malignancies such as acute myeloid leukemia, multiple myeloma, and non-Hodgkin lymphoma and Hodgkin lymphoma. As of November 2023, seven clinically approved TCE therapies are on the market. TCE-based therapies still have their limitations; however, improving the properties of TCEs, as well as combining TCE-based therapies with other forms of treatment, give hope to find the cures for currently terminal diseases. In this paper, we summarized the technical basis of the TCE technology, its application in hematology, and its current issues and prospects.

## 1. Introduction

Recently, directing the cellular immune response onto cancer cells has become a promising strategy for the treatment of hematological malignancies. Although the introduction of monoclonal antibody-based (mAbs) targeted therapy has significantly improved the prognosis for hematological patients, this type of treatment still has its limitations. The long-term efficacy of mAbs is restricted by the mechanisms of drug resistance [1]. Additionally, this type of agent does not activate the response of cytotoxic T cells, which have the biggest contribution in the immune response towards cancer cells [2].

In response to these issues, a novel bispecific antibody (BsAb) format was developed. This type of molecule can bind to multiple antigens, which allows for several brand-new applications, such as directing immune activity onto target cancer cells. Those advancements reduce the occurrence of severe adverse events and prevent the development of drug resistance [3]. The original concept of BsAb was first proposed by Nisonoff and his collaborators in the 1960s [4,5]. Since then, numerous studies providing insight into antibody architecture led to the invention of hybridoma technology in 1975. This discovery solved the problem of producing large quantities of pure antibodies, which then allowed the development of a new kind of therapies that utilize mAbs [6]. Another breakthrough discovery was made in 1983, when Milstein and Cuello pushed further the idea of hybridoma lines, which resulted in the creation of the hybrid-hybridoma (quadroma) technology that allowed the production of the first BsAbs [7]. Shortly after, in 1988, it was followed by the Huston team inventing a novel antibody-based protein molecule—the single-chain variable fragment (scFv). This achievement has greatly minimized production errors, such as refolding problems, mainly incorrect domain pairing or aggregation of two-chain species [8]. However, it was the knobs-into-holes (KiH) technology developed in 1996 that allowed for the construction of many BsAbs that we know today [9]. Establishing the novel format of BsAbs caused a series of experiments regarding the choice of target antigens, which eventually led to the creation of T-cell engagers (TCEs). TCEs are a broad family of agents which share a key common feature. They all simultaneously target some type of tumor-associated antigen (TAA) and a CD3 molecule present on T-cells, which allows for directing T-cells’ cytotoxic activity against cancer cells.

The bispecific T-cell engager (BiTE; Micromet, Munich, Germany) molecules have been typed as one of the most promising agents of this kind. In 2014, the FDA approved blinatumomab, a BiTEtargeting CD19 and CD3, in the treatment of relapsed or refractory (r/r) B-cell precursor acute lymphoblastic leukemia (B-ALL) [10]. Since then, TCEs have been a popular study topic which has led to the development of various molecules targeting different tumor antigens [11]. In this paper, we aimed to present the state of the art of TCEs, including the general concept, production platforms and mechanism of action, as well as their application in hematology, current limitations and prospects.

### 1.1. TCEs as Members of BsAbs Family

According to their targets, BsAbs can be divided into three major groups. The first group includes molecules targeting two various TAAs. This technique can help to avoid the destruction of physiological cells expressing just one TAA, thus providing more precise tumor cell targeting in therapy.

TG-1801 is an example of this type of BsAb. Simultaneously targeting CD19, naturally occurring on B-cell lineage and CD47, commonly expressed by tumor cells as a means to avoid macrophage phagocytosis, it destroys tumor cells in B-cell lymphoma, reducing the killing of regular B cells [12]. Despite obvious advantages, this type of agent does not activate the T-cell activity, which has the biggest contribution in anti-tumor response. BsAbs classified in the second group target two immune-related molecules. This type of antibody can be used for relieving an immunosuppressive phenotype. Most of these BsAbs are a combination of programmed cell death protein 1 (PD-1) and other immune checkpoint inhibitors. Previous data have shown that PD-1 × CTLA-4 (cytotoxic T-lymphocyte associated protein 4) (MEDI5752) and PD-1 × lymphocyte-activation gene 3 (LAG-3) (YG-003D3) BsAbs have great potential for clinical application [13,14]. The molecules of the third group are the combination of the two aforementioned classes. They target both TAAs and immune-related antigens. TCEs are a prime example of antibodies from this class, as they target one CD3 molecule and one TAA simultaneously.

Another factor used to divide BsAbs as well as TCEs is the molecular structure of those agents. According to this criterion, BsAbs can, once again, be divided into three groups. The first group includes the immunoglobulin G-based antibodies (IgG-based/IgG-like), BsAbs that are developed using the blueprint already present naturally in mAbs, thus resembling native antibodies. The second group, termed variable fragment-based antibodies (Fv-based/non-IgG-like) comprises molecules mainly composed of synthetic scFvs. These agents greatly differ from natural antibody constructs, both in structure and physiological properties. The last group is a hybrid of the previous two and includes agents combining the features of IgG-like and Fv-based BsAbs, such as the presence of both fragment crystallizable region (Fc) domains and scFv molecules. The general construction of TCE molecules is presented in Figure 1.

#### 1.1.1. IgG-like TCEs

As previously mentioned, IgG-like TCEs are developed based on native antibodies, namely, the IgG molecules. This means that, similarly to natural antibodies, they are composed of two heavy and two light chains. In their structure, we can distinguish the Fc, as well as, connected to it by A hinge, the two antigen-binding regions (Fab), featuring a Fv on each. Contrary to native mAbs, in which both Fvs bind to the same type of antigen, in IgG-like TCEs, one Fv binds to a TAA and the other to a CD3 molecule [10]. The presence of the Fc domain is the most significant difference between IgG-like and non-IgG-like TCEs. Thanks to this feature, compared with Fv-based TCEs, IgG-based TCE molecules are larger, thus harder to be cleared by the kidney. This, in turn, grants IgG-based BsAbs with a longer half-life in vivo, ranging up to 7 days. The presence of the Fc domain also improves the solubility and stability of IgG-like BsAbs [15]. Apart from influencing the physical properties, Fc domains of BsAbs can take part in the immune response. They can recruit natural killer (NK) cells and macrophages to induce antibody-dependent cell-mediated cytotoxicity (ADCC) and complement-dependent cytotoxicity (CDC) [16].

Despite obvious benefits, there are significant disadvantages stemming from the presence of the Fc domains. Compared to Fv-based BsAbs, the larger molecular weight of IgG-based BsAbs greatly decreases their tumor tissue permeability. Additionally, obtaining IgG-based antibodies requires more complex techniques. In the early days of BsAb development, BsAbs were produced by the reduction and reoxidation of hinged cysteine in monoclonal antibodies targeting two different antigens, synthesized in hybridoma cell lines [17]. This technique has proven faulty, as the heavy and light chains, when mismatched, may produce a variety of side products [18,19,20]. Combating this phenomenon, in many IgG-like BsAb platforms, the heavy chain has been modified to promote heterologous Fc matching, including the KiH technique, in which the local spatial structure of Fc is changed [19,21]. Based on this concept, additional exchange of fragments in heavy and light chains led to the development of CrossMab technology. Another popular IgG-like platform is the DuoBody, based on the controlled Fab dynamic recombination exchange [22].

##### Knobs-into-Holes (KiH)

KiH technology developed by Roche enables the production of antibodies through exchanging half-molecules. This technique took its name from the mechanism used to promote the dimerization of two halves of antibodies targeting different antigens. The principle behind KiH is based on creating puzzle-like matching sites on two different antibody heavy chains meant to create the product. It is done by replacing a smaller amino acid with a larger amino acid (T336Y) in the CH3 region of an antibody chain to form a “knobs” structure, and at the same time substituting a larger amino acid in the other chain with a smaller amino acid to form a “holes” structure (Y407T). This technique grants a recombination efficiency of 57% [21]. The KiH platform does not solve the problem of mismatching of the heavy and light chains. Facing this obstacle, Merchant et al. developed a tactic for constructing human IgG-like BsAbs that greatly eliminates mispairing between light chains and heavy chains, improving the heterodimerization ratio up to a maximum of approximately 95% [9].

##### CrossMab

Based on the original KiH concept, Roche developed the CrossMab platform with the premise of solving the problem of light chain mismatching. Similarly to the regions on heavy chains in KiH, this technique is based on swapping the regions of one side’s heavy chain and light chain, so that the heavy and light chains can be assembled correctly. Various regions of the heavy and light chains can be modified, although it has been shown that exchanging the CH1 of the heavy chain with the CL of the light chain is the most effective method; hence, it is currently used as the most common method [23,24].

With the utilization of the CrossMab platform, in addition to bivalent BsAbs structures, multivalent BsAbs can also be generated [25]. Vu et al. were one of the first to report a promising activity of an agent in a 2:1 format, bivalent to BCMA (B-cell maturation antigen), monovalent to CD3 [26].

##### DuoBody

Duobody developed by Genmab is the platform that enables production of BsAbs by exchanging Fab regions between two different IgGs. The mutation in the CH3 region of the Fc fragment can recognize the heterologous half-molecule and promote the procedure of heterodimerization. The core of the Duobody technology lies in the process known as controlled Fab-arm exchange (cFAE). The method involves the separate expression of two different kinds of IgG mAbs that each feature a single matched point mutation at the CH3–CH3 domain interface. During the controlled reduction of hinge disulfide bridges in vitro, the matched mutations allow for the efficient recombination of binding arms, resulting in the production of asymmetric BsAbs [27].

#### 1.1.2. Fv-Based TCEs

Compared to IgG-based BsAbs, the design of Fv-based TCEs is relatively simple. These molecules do not resemble naturally occurring antibodies, as they are usually composed of scFvs only. The scFv bears similarities with the Fv region of the antibody both in structure and function. It is a product of the artificial fusion of one variable region of the heavy chain (VH) and one variable region of the light chain (VL) [8]. Due to the lack of Fc domains, Fv-like TCEs present therapeutic effects simply through antigen binding. Additionally, due to their small size, they allow the formation of a short and stable cytolytic synapse. They are easy to produce and have low immunogenicity [28]. Although Fv-based TCEs have high tumor tissue permeability because of their low molecular weight, compared to IgG-based TCEs, they have a short half-life and require multiple doses [28,29,30].

The BiTE technique is the prime platform to produce Fv-based BsAbs. Other examples of common non-IgG-like TCEs include dual-affinity retargeting antibody (DART), and tandem diabody (TandAb) [31,32,33].

##### Bi-Specific T-Cell Engagers (BiTE)

A BiTE is a simple molecule that connects the CD3-specific ScFv and the TAA-specific ScFv with G4S linker [34,35]. Apart from blinatumomab, other examples of BsAbs that use the BiTE platform include CD3 × BCMA-AMG420 and CD3 × CD33-AMG330, currently evaluated in clinical trials (NCT03836053, NCT02520427) [36,37,38,39]. BsAbs on the BiTE platform are small and have a short half-life of only about 2 h, which means administration with continuous intravenous infusion is required [40,41].

##### Dual-Affinity Retargeting Antibody (DART)

DART is a platform similar to BiTE. It is formed by linking VH and VL sequences with other antibody VL and VH sequences, respectively. Additionally, cysteine is introduced at the C-terminus of the two polypeptide chains to form an interchain disulfide bond. MGD024 is an example of TCE created by Macrogenics on the DART platform. It targets CD3 and CD123 antigens and is studied for application in r/r CD123-positive malignancies. This agent was engineered to lower the occurrence of cytokine release syndrome (CRS), which was a common AE in its predecessor—flotetuzumab, while maintaining antitumor activity. It is currently being evaluated in a phase 1, first-in-human, dose-escalation study (CP-MGD024-01; NCT05362773) [42,43]. In comparison with the BiTE designs, DART achieves a higher magnitude of T cell activation [44].

##### TandAb

The TandAbs platform is a tetravalent antibody molecule with two binding sites for each of two antigens. A homodimer molecule is formed by the reverse pairing of two peptide chains [45]. AFM11, which targets CD3 and CD19, is a BsAb developed by Affimed on the TandAb platform. The safety and efficacy of AFM11 have been evaluated in two open-label, multicenter, dose-escalation phase 1 studies, in patients with r/r CD19-positive B-cell non-Hodgkin lymphoma (NHL) (AFM11-101) and in patients with CD19-positive B-precursor Philadelphia-chromosome-negative (Ph-) ALL (AFM11-102). Neurological adverse reactions, severe in some patients, were the most common, as well as dose-limiting treatment-emergent adverse events (TEAEs). In ALL, albeit short-lived, the drug has shown signs of activity, whereas no activity was observed in patients with NHL. Due to major health risks associated with the AFM11 treatment, further clinical development was terminated [46].

#### 1.1.3. Combination-Based TCEs

Combination-type TCEs use the benefits of platforms from both IgG-like and Fv-like groups. Those agents feature both the Fc region characteristic for the IgG-like TCEs and Fcvs typical for Fv-like TCEs. HLE-BiTE is a leading example of combination-type TCE. It is a second-generation BiTE, developed by Amgen by connecting the Fc fragment to the end of the ScFv of the original construct. This modification extends its half-life to 7 days [47]. It is applied in Amgen’s AMG701 BsAb, an antibody targeting CD3 and BCMA. A phase 1 first-in-human study of (NCT03287908) demonstrated a manageable safety profile, encouraging activity, and a favorable pharmacokinetics profile in patients with heavily pre-treated r/r MM, supporting further evaluation of AMG 701 [48]. Another example from this group is the XmAb platform, also developed by Amgen, this time in collaboration with Xencor. This type of molecule features both natural Fab regions and synthetic scFv connected to the hetero-Fc domain. Plamotamab is a BsAb developed on the XmAb platform. Its safety and efficacy are currently evaluated in a first-in-human, multi-center, open-label phase 1 dose-escalation study in r/r NHL patients (NCT02924402). So far, plamotamab has demonstrated evidence of clinical activity in heavily pretreated patients with diffuse large B-cell lymphoma (DLBCL) and follicular lymphoma (FL) with promising responses in patients with prior chimeric antigen receptor (CAR)-T therapy. CRS was generally manageable with premedication [49]. All three types of TCE, along with the examples from each group, are presented in Figure 2.

## 2. Mechanism of TCEs’ Anti-Tumor Action

The mechanism of TCE action against tumor cells is highly dependent on the molecular structure of the TCE agents themselves. It can be generally categorized into two different pathways, according to the region of the molecule contributing to the antitumor response. The first pathway is ubiquitous to all TCE platforms and, in fact, constitutes their membership in this agent family. It is based on the ability of simultaneous binding to TAA and CD3, which leads to the immediate connection of tumor cell and cytotoxic T-cell, ultimately resulting in the lysis of the tumor cell. The second pathway is restricted to limited types of IgG-like TCEs and involves Fc-dependent mechanisms of immunological response. The overall scheme of TCE mechanism of action is represented in Figure 3.

### 2.1. TCE Action through Antigen Binding

The advantage of TCE over natural antibodies stems from the ability to redirect T cells to specific tumor antigens and activate T cells directly. Natural antibodies are unable to recruit T cells, as these lack Fcγ receptors, responsive to the antibody Fc domain. Unique properties of the BiTE molecule allow its simultaneous binding to CD3 on the T cell and TAA on the cancer cell [50]. The CD3 molecule non-covalently associates with the T-cell receptor (TCR) and participates in antigen-specific signal transduction which can induce the activation of T cells. Importantly, T-cell activation does not occur if BiTE binds only to the CD3 [51,52]. Establishing the CD3-BiTE-TAA complex allows the formation of the cytolytic synapse, which leads to the T cell releasing perforins and cytotoxic granzyme-B, which then results in the lysis of the target cancer cell [51].

The activity of TCEs stimulates the polyclonal T-cell activation, inducing a robust proliferation of the T-cell compartment. Stimulation by BiTE promotes the expression of CD69 and IL2RA, known as T-cell activation markers, on the vast majority of CD8 and CD4 T-cells, thus allowing all CD8 and CD4 T-cell subpopulations except the naive T cells to be involved in redirected tumor cell lysis [53]. Additionally, T cells activated by TCEs secrete a wide range of cytokines, including IL-2, IFN-γ (interferon γ) and TNF-α (tumor necrosis factor α), which enhance their anti-tumor effector function [54]. Importantly, a strong T-cell response can occur even against tumors that do not express major histocompatibility complex (MHC) class I molecules. Loss of human leukocyte antigen (HLA) expression is well known, and is one of the most common mechanisms of tumor immune evasion [55].

The activation of T cells without co-stimulation could be explained by two theories. One of them suggests that tumor cells express ICOS (inducible T-cell costimulator) ligand and CD80/CD86 on their surface in amounts sufficient for co-signaling of the CTL through the CD28 costimulatory family [56]. The other one proposes a model in which tumor lysis by TCEs is mainly mediated by memory T cells, which do not require the second signal for activation. Hence, CD28 activity, instead of inducing a unique array of signaling pathways, may just enable the direct T-cell response. Therefore, if the signaling thresholds vary among different memory T-cell populations, in some cases, inducing the TCR-CD3 pathway via TCEs may be sufficient to drive activation without costimulation [54,57].

Additionally, it has been also shown that TCEs can revive dysfunctional T cells, which could be then used to improve the effectiveness of antigen-exhausted tumor-infiltrating lymphocytes in approaches of cellular therapy [58].

### 2.2. Fc-Dependent TCE Action

IgG-like BsAbs can retain Fc-mediated effector functions such as ADCC, CDC, and antibody-dependent cellular phagocytosis. Catumaxomab, an anti-epithelial cell adhesion molecule (EpCAM)-anti-CD3 BsAb, is an example of TCE with a functional Fc. Catumaxomab was designed to bind the Fcγ receptor, thus activating the ADCC mechanism [59]. However, it was found that this feature greatly increased the risk of dose-limiting toxicities, namely the CRS [60].

Nowadays, for most TCEs, the Fc domain is silenced, as Fc-mediated immune functions are not necessary for inducing effective T-cell response [61]. To avoid or reduce CRS due to crosslinking of CD3 and Fcγ receptors, Fcγ receptors are “switched off” via introduced mutations which eliminate FcγR binding [62].

## 3. TCEs in the Treatment of Hematological Malignancies

Over the last few decades, the knowledge regarding anti-tumor immune response has significantly expanded, which has increased the interest in developing cancer immunotherapies. The unique properties of TCE technology enable its application in novel cancer therapies. Most approved BiTEs are used in the treatment of hematological malignancies. While TAAs have been identified both in hematological malignancies and solid tumors, most are also present in normal cellular counterparts, which results in “on-target/off-tumor” toxicities. These on-target toxicities in normal tissues are more manageable in hematological malignancies than in solid tumors. For example, CD19-targeting results in B-cell aplasia and hypogammaglobulinemia and increases the risk of infection; such infections can be managed by being vigilant clinically and the prompt use of antibiotics. On the other hand, targeting epidermal growth factor (EGFR) in lung cancer leads to generalized cutaneous toxicity and cardiotoxicity in the treatment of human epidermal growth factor receptor 2 (HER2)-positive breast cancer using trastuzumab [63,64].

In 2014, the FDA approved blinatumomab, a TCE directed to CD3 and CD19 to treat acute lymphoblastic leukemia (ALL). Nevertheless, the approved indication concerns three different settings of ALL: positive minimal residual disease (MRD), r/r Ph+, and r/r Ph− in both pediatric and adult cases [65]. The success of blinatumomab resulted in the development of TCEs targeting antigens other than CD19, such as CD20, B-cell maturation antigen (BCMA) PI3, CD33, FMS-like tyrosine kinase 3 (FLT3), and more. This, in turn, created the possibility of treating different hematological malignancies, like acute myeloid leukemia (AML), multiple myeloma (MM), and NHL and Hodgkin lymphoma (HL).

### 3.1. Antigens Frequently Targeted in Hematology

#### 3.1.1. CD19

CD19 is a molecular marker ubiquitously expressed on the surface of B cells [66]. Along with CD79a, CD79b, and cytoplasmic signaling and accessory molecules, it creates the B-cell receptor (BCR) complex [67]. The functional role of CD19 is to decrease the threshold for the activation of B cells mediated by the BCR [68]. Because CD19 is broadly and consistently expressed throughout B-cell development, it is an attractive target across all B-cell malignancies; however, this feature also leads to the treatment-related depletion of physiological B cells, which results in AEs during immunotherapy [66].

Blinatumomab is a prime example of TCE targeting CD19. Apart from B-ALL, it is being investigated in additional B-cell malignancies, including NHL, as both monotherapy and combination therapy (eg, NCT03114865, NCT02910063, and NCT03072771) [69,70,71]. As an alternative to continuous intravenous dosing, subcutaneous delivery is being investigated in a phase 1b study (NCT02961881) [72]. Additionally, AMG 562, an HLE-BiTE molecule is currently being evaluated in a first-in-human study in patients with r/r DLBCL, mantle cell lymphoma, and FL is recruiting (NCT03571828) [73].

#### 3.1.2. CD20

CD20 is another antigen typical for the B-cell lineage, although it is expressed in a developmentally restricted manner. It is initiated at the pre-B-cell stage of development and remains present until terminal differentiation into a plasma cell [74]. The biological activity of CD20 is not fully determined, however, it is thought to act as an ion channel and a store-operated Ca2+ channel [75]. CD20 is also thought to function as a modulator of cell growth and differentiation, and to initiate intracellular signals.

CD20 is a popular target in TCE-based therapies; in fact, most approved TCE therapies, such as mosunetuzumab, glofitamab, and epcoritamab target this antigen. Apart from already approved agents, odronextamab (REGN1979), a CD20 x CD3-targeting IgG-like BsAb developed by Regeneron, is currently being evaluated for the treatment of CD20-positive B-cell malignancies, such as DLBCL and FL (NCT02290951, NCT03888105) [76]. Recently, on 17 August 2023, the EMA accepted for review the Marketing Authorization Application (MAA) for odronextamab to treat adult patients with r/r FL or r/r DLBCL, who have progressed after at least two prior systemic therapies [77].

#### 3.1.3. B-Cell Maturation Antigen (BCMA)

BCMA, also referred to as TNFRSF17 or CD269, is a member of the tumor necrosis factor receptor (TNFR) superfamily. Ligands for BCMA include B-cell activating factor (BAFF) and a proliferation-inducing ligand (APRIL) [78]. Physiologically, BCMA is expressed mainly by mature B lymphocytes, with minimal expression in hematopoietic stem cells or nonhematopoietic tissue. The overexpression and activation of BCMA are associated with the progression of MM, making BCMA a promising target in the treatment of MM [79].

There are currently two clinically available anti-CD3-anti-BCMA therapies—teclistamab and elranatamab. Additionally, Amgen is developing AMG 420, a BiTE molecule that, in preclinical studies, triggered the lysis of BCMA-expressing cells. The AMG 420 first-in-human phase 1 dose-escalation study treated patients with r/r MM who had received ≥2 prior treatment lines. The maximum tolerated dose was 400 µg daily; at that dose, the overall response rate (ORR) was 70% (7 of 10 patients), and 5 of the 7 patients achieved an MRD-negative CR. Grade 3 peripheral polyneuropathy was a dose-limiting toxicity in 1 patient (2.5%), but has been resolved with intravenous tocilizumab and corticosteroid administration (NCT02514239) [80].

#### 3.1.4. CD33

CD33 is a member of the sialic acid binding immunoglobulin-like lectin family. CD33 is expressed on all normal myeloid cells derived from the common myeloid progenitor and is used as both a diagnostic marker and a therapeutic target for AML, myelodysplastic syndrome (MDS), and chronic myeloid leukemia (CML) [81].

There are currently no CD3 × CD33 TCEs approved for clinical use. AMG 330, a BsAb developed by Amgen on the BiTE platform, has been evaluated in a phase 1 dose-escalation study of AMG 330 in patients with r/r AML for the safety, pharmacokinetics, pharmacodynamics, and maximum tolerated dose. Preliminary data are encouraging; AMG 330 dosed at up to 480 µg daily is tolerable and has antileukemic activity in heavily pretreated patients (NCT02520427) [39].

#### 3.1.5. FMS-like Tyrosine Kinase 3 (FLT3)

The FLT3 antigen has been detected in most AML blasts and leukemic stem cells, whereas cell surface expression on nonmalignant cells is limited to immature hematopoietic progenitor cells. Activating mutations in FLT3 account for 30% of all AML cases and are responsible for increased cell proliferation and decreased cell apoptosis [82,83].

Similarly to CD33, there are no clinically approved anti-FLT3 TCEs yet. CLN-049 is an example of currently developed anti-CD3-anti-FLT3 BsAbs. It is designed by Cullinan Oncology as an IgG heavy chain/scFv fusion. Its safety and efficacy are currently being evaluated in a phase 1, open-label study in patients with r/r AML and MDS (NCT05143996) [84].

## 4. Currently Approved TCE Therapies in Hematology (November 2023)

As of November 2023, seven clinically approved TCE therapies are on the market, all of which are used to treat lymphoproliferative diseases.

The first-ever TCE therapy to be allowed for clinical use was blinatumomab (Blincyto, Amgen, Thousand Oaks, CA, USA). It was approved in 2014 by the FDA to treat r/r B-ALL. Later in 2015, it was approved by the EMA.

Blinatumomab remained the only available TCE-based therapy for many years, up to 8 June 2022, when Roche announced that the European Commission had granted conditional marketing authorization for the CD20 x CD3 TCE Lunsumio (mosunetuzumab) for the treatment of adult patients with r/r FL, who have received at least two prior systemic therapies.

Soon after that, on 24 August 2022, The Janssen Pharmaceutical Companies of Johnson & Johnson announced that the European Commission (EC) had granted conditional marketing authorization (CMA) of TECVAYLI (teclistamab) as monotherapy for the treatment of adult patients with r/r MM. On 25 October 2022, the FDA granted accelerated approval to TECVAYLI.

Nowadays, the development of new TCE therapies has accelerated, with four new therapies already approved in 2023. On March 24, Hoffmann-La Roche Limited (Roche Canada, Mississauga, ON, Canada) announced that Health Canada authorized COLUMVI (glofitamab) to treat adult patients with r/r DLBCL.

Another agent used for the treatment of DLBCL is epcoritamab-bysp (Epkinly, Genmab US, Inc., Plainsboro, NJ, USA) which received accelerated approval from the FDA on 19 May 2023.

Lastly, two approved agents are used to treat r/r MM, the first of which, TALVEY (talquetamab-tgvs), is the first BsAb to target CD3 and the G protein–coupled receptor, family C, group 5, member D (GPRC5D). On 10 August 2023, it received accelerated FDA approval for the treatment of adult patients who have received at least four prior lines of therapy, including a proteasome inhibitor, an immunomodulatory agent, and an anti-CD38 antibody.

Most recently, on 14 August 2023, the FDA granted accelerated approval to elranatamab-bcmm (Elrexfio, Pfizer, New York, NY, USA) for the treatment of adults with r/r MM. All currently approved TCE-based therapies are summarized in Table 1.

### 4.1. Blinatumomab

Blinatumomab (AMG103) is a BiTE molecule created by Amgen. It is the first TCE to be approved for clinical use, and is, to date, the only approved Fv-based TCE.

After confirmation of blinatumomab’s cytotoxicity against CD19+ B cells in preclinical studies, BiTE was administered to ALL patients in complete response (CR), but with positive MRD+ status [85]. Of the patients receiving the drug, 81% achieved MRD- status with a relapse-free survival rate (RFS) of 61% after a median observation time of 33 months [54]. This pilot study was followed up by a multicenter, single-arm, phase 2 trial on the efficacy and tolerability of blinatumomab in ALL, MRD+ patients. The results showed that 78% of 113 evaluated patients achieved MRD- status with a 54% RFS rate at 18 months and with a median overall survival (OS) rate of 36.5 months. Compared to patients with persistent MRD status, complete MRD responders had higher RFS and OS rates (respectively, 23.6 vs. 5.7 months; *p* = 0.002, 38.9 vs. 12.5 months; *p* = 0.002) [85].

To investigate the blinatumomab’s potential in the treatment of ALL in different settings, further studies concerning ALL r/r Ph- and r/r Ph+ patients were performed. After the second cycle of treatment, 33% of 189 patients with r/r Ph- ALL achieved CR and 10% had CR with partial hematological recovery (CRh). The median RFS and OS were 5.9 months and 6.1 months, respectively [86]. Confirmation of blinatumomab’s antileukemic properties resulted in a multicenter phase 3 trial (TOWER), in which the efficacy of this novel drug was compared with the standard chemotherapy. The data from this study revealed that CRh and CR rates were higher in patients treated with blinatumomab than in patients with standard chemotherapy treatment (44% vs. 25%, *p* < 0.001, 34% vs. 16%, *p* < 0.001, respectively). Additionally, the OS significantly increased in the group treated with blinatumomab (7.7 months vs. 4.0 months) as well as the median CR duration (7.3 vs. 4.6 months) [87].

The ALL r/r Ph+ patients in whom imatinib treatment was not effective or not possible, as well as patients who were intolerant to at least the first generation of tyrosine kinase inhibitors (TKIs), were administered blinatumomab. Of the 45 patients, 88% achieved complete MRD-response, and the median RFS and OS ranged 6.7 months and 7.1 months, respectively [88].

The most common side effects of blinatumomab treatment include pyrexia, headache, leukopenia, lymphopenia, rigor, hypokalemia, anemia, and constipation. The majority of blinatumomab’s AEs appear during the first cycle in mild to moderate intensity. CRS is not frequent. The treatment-related mortality is low, although so far, there is no evidence of a possible connection to the drug’s influence on this issue [85,86].

Most of the ongoing studies of blinatumomab in ALL focus on the efficacy of the first-line BiTE therapy combined with various chemotherapy regimens. Studies for CLL/NHL are also currently running. They include those for MRD treatment in DLBC NHL after autologous hematopoietic stem cell transplantation (HSCT) (NCT03298412), r/r indolent NHL as subcutaneous formulation (NCT02961881), Richter transformation (NCT03121534, NCT03072771), r/r indolent or aggressive NHL (NCT02811679, NCT02910063), first line in DLBC NHL (NCT03023878), and in combination with lenalidomide in r/r NHL (NCT02568553). Additionally, the research for blinatumomab in r/r MM treatment is also being performed (NCT03173430) [85]. The best responses for blinatumomab were observed in patients with FL. The ORR was 80%, and CR was achieved by 40% of patients. Moreover, the data from a phase 2 clinical trial for r/r DLBCL showed that the ORR was 43%, with CR in 19% of patients and 3.7 months of progression-free survival (PFS). Nevertheless, the trials concerning AEs and tolerability need to be performed [89]. Additionally, blinatumomab’s activity against CLL cells was proven in the in vitro studies in 2003; however, the amount of evidence regarding its clinical activity is still not sufficient [90,91].

### 4.2. Mosunetuzumab

Mosunetuzumab (RG7828), created by Roche via KiH technology, is a humanized bispecific anti-CD20-anti-CD3 IgG-like antibody with a full-length Fc region providing a longer half-life in the human organism. Additionally, the amino acid substitution in this Fc region prevents activation of ADCC, which potentially lowers the risk of therapy-related cytotoxicity [92].

The safety and efficacy of mosunetuzumab in patients with r/r FL is investigated in an ongoing, single-arm, multicenter, phase 2 study (NCT02500407). Between 2 May 2019 and 25 September 2020, 90 patients who had received two or more previous therapies were enrolled. At the time of the data cutoff (27 August 2021), the median follow-up was 18.3 months. CR was recorded in 54 patients (60.0%), which was significantly higher than the control CR rate achieved with a phosphoinositide 3-kinase (PI3K) inhibitor, copanlisib of 14% [93].

CRS was the most common AE (44% of patients) and was predominantly grade 1 (26.7%) and grade 2 (17.3%), and primarily occurred in the first cycle of the treatment. The most common grade 3–4 AEs were neutropenia (26.7%), hypophosphatemia (16.7%), hyperglycemia (7.8%), and anemia (7.8%). Serious AEs occurred in 42 (46.7%) of 90 patients. No treatment-related fatal events occurred [94].

Mosunetuzumab was conditionally approved in the EU for the treatment of r/r FL in adults who have received at least two prior systemic therapies [95].

### 4.3. Teclistamab

Teclistamab (JNJ-7957) is a humanized BsAb developed by Janssen on the Duobody platform, targeting BCMA and CD3 [96].

It has been clinically proven that utilizing teclistamab results in a high rate of deep and durable response in patients with r/r MM. A phase 1–2 study (NCT03145181 and NCT04557098) involved 165 r/r MM patients after at least three therapy lines (median, five previous therapy lines), including triple-class exposure to an immunomodulatory drug, a proteasome inhibitor, and an anti-CD38 antibody. With a median follow-up of 14.1 months, the ORR was 63.0%, with 65 patients (39.4%) having a CR or better. A total of 44 patients (26.7%) were found to be MRD-negative. The median duration of response was 18.4 months. The median duration of progression-free survival was 11.3 months. Common AEs included CRS (72.1%, including grade 3, 0.6%), neutropenia (in 70.9%), anemia (52.1%), and thrombocytopenia (40.0%). Infections were frequent (76.4%). Neurotoxic events occurred in 24 patients (14.5%), including immune effector cell-associated neurotoxicity syndrome (ICANS) in 5 patients (3.0%) [97]. Teclistamab has received conditional approval in the EU for the treatment of adult patients with r/r MM who have received three or more prior therapies (including an immunomodulatory agent, a proteasome inhibitor, and an anti-CD38 antibody) and have demonstrated disease progression on the last therapy. Teclistamab was subsequently approved in the US for the treatment of adult patients with r/r MM who have received at least four prior lines of therapy (including an immunomodulatory agent, a proteasome inhibitor, and an anti-CD38 antibody) [98].

### 4.4. Glofitamab

Glofitamab (RG6026) is a bispecific anti-CD20-anti-CD3 antibody. It was created by Roche with the use of Crossmab technology in a novel 2:1 format. This unique molecular configuration has two CD20 binding Fab regions, providing a higher affinity to a CD20 antigen and one scFv CD3 binding site. In addition, it also has a modified Fc region with completely suppressed binding to FcγRs and C1q, which contributes to reduced toxicity and an extended half-life [99].

Glofitamab is an agent effective for the DLBCL therapy. The phase 2 part of a phase 1–2 study (NCT03075696) was performed on the cohort of 155 patients who had previously received at least two lines of therapy. Before the administration of glofitamab monotherapy, patients were pretreated with a monoclonal anti-CD20 antibody, obinutuzumab, to mitigate CRS. Among the enrolled patients, 154 received at least one dose of any study treatment (obinutuzumab or glofitamab). At a median follow-up of 12.6 months, 39.4% of the patients had a CR according to an independent review. The median time to a CR was 42 days. The majority (78.0%) of CR were ongoing at 12 months. The 12-month progression-free survival was 37.0%.

More than half the patients presented AEs of grade 3 or 4. The administration of glofitamab had to be ceased due to AEs in 9.1% of the patients. The most common AE was CRS (in 63.0% of the patients, including 4% of grade 3 or higher). ICANS occurred in 12 patients (8.0%), with events of grade 3 or higher in 3.0% [100].

Glofitamab received its first approval (with conditions) on 25 March 2023, in Canada, for the treatment of adult patients with r/r DLBCL not otherwise specified, DLBCL arising from FL, or primary mediastinal B-cell lymphoma, who have received two or more lines of systemic therapy and are ineligible to receive or cannot receive CAR T-cell therapy or have previously received CAR T-cell therapy. Glofitamab is also under regulatory review for r/r DLBCL in the EU and USA and, in April 2023, received a positive opinion recommending the granting of a conditional marketing authorization in the EU. Clinical development of glofitamab, as a monotherapy and in combination with other agents for the treatment of NHL, is continuing worldwide [101].

### 4.5. Epcoritamab

Epcoritamab (GEN3013) is a product developed by Genmab using the Duobody technology. It is a subcutaneously administered, bispecific IgG-like antibody targeting CD3 and CD20 antigens. The Fc region of DuoBody-CD3 × CD20 is silenced by three point mutations that were selected based on functional assays [102].

The safety and efficacy of epcoritamab is currently being evaluated in an ongoing dose-expansion cohort of a phase 1–2 study (NCT03625037). As of 31 January 2022, the study included 157 patients aged from 20 to 83 years, who were previously treated with at least two therapy lines (medium three), including CAR-T therapy (38.9%), for r/r DLBCL and other aggressive forms of NHL. At a median follow-up of 10.7 months, the ORR was 63.1% and the CRR was 38.9%. The median duration of response was 12.0 months. The most common AEs were CRS (49.7%, including grade 3: 2.5%), pyrexia (23.6%), and fatigue (22.9%). ICANS occurred in 6.4% of patients, with one fatal event [103].

Epcoritamab received its first (conditional) approval on 19 May 2023, in the USA, for the treatment of adult patients with r/r DLBCL, ≥2 lines of systemic therapy. Similarly, in the EU, epcoritamab was granted approval as a monotherapy for the treatment of adults with r/r DLBCL after ≥2 lines of systemic therapy. Additionally, it is currently under regulatory review in Japan for the treatment of adults with r/r large B-cell lymphoma after ≥2 lines of systemic therapy. Clinical development of epcoritamab as monotherapy and in combination with standard of care agents for the treatment of mature B-NHLs is ongoing globally [104].

### 4.6. Talquetamab

Talquetamab is an IgG-like BsAb developed by Janssen on the Duobody platform. It is the first clinically available anti-CD3-anti-G-protein coupled receptor family C group 5 member D (GPRC5D) TCE. GPRC5D is an orphan receptor that is primarily present in plasma cells and hard keratinized tissues, including cortical cells of the hair shaft, the keratogenous zone of the nail, and in a central region of the filiform papillae of the tongue, with low expression in normal human tissues [105]. GPRC5D overexpression has been detected in the bone marrow of patients with MM and correlates positively with a high plasma cell count, making it a suitable marker for the treatment of MM [106].

The efficacy of talquetamab was evaluated in MMY1001 (MonumenTAL-1; NCT03399799, NCT4634552), a single-arm, open-label, multicenter study, that included patients diagnosed with r/r MM who had previously received at least four prior systemic therapies. At the data-cutoff date, 232 patients had received talquetamab, including 102 intravenously and 130 subcutaneously at two doses recommended for a phase 2 study (405 μg/kg weekly [30 patients] and 800 μg/kg every other week [44 patients]. At median follow-ups of 11.7 months (in 405-μg dose level) and 4.2 months (in 800-μg dose level), the percentages of patients with a CR or better reached around 23.0% in both groups. The median duration of response was 10.2 months and 7.8 months, respectively.

CRS was the most common AE (in 76.7% and 79.5% of the patients, respectively), with one grade 4 event. Other AEs include skin-related events (in 66.7% and 70.5%), and dysgeusia (in 63.3% and 56.8%). One dose-limiting toxic effect of grade 3 rash was reported in a patient who had received talquetamab at the 800-μg dose level [107].

The FDA has granted accelerated approval to talquetamab-tgvs (Talvey) for the treatment of adult patients with r/r MM who have received at least 4 prior lines of therapy, including a proteasome inhibitor, an immunomodulatory agent, and an anti-CD38 antibody. The European Commission (EC) has granted conditional marketing authorization to talquetamab-tgvs (Talvey) monotherapy for the treatment of patients with r/r MM who have received at least 3 prior therapies, including an immunomodulatory agent, a proteasome inhibitor, and an anti-CD38 antibody, and have demonstrated disease progression on the last therapy [108].

### 4.7. Elranatamab

Elranatamab (PF-06863135), is a novel, humanized full-length bispecific IgG-like antibody derived from two mAbs, the anti-BCMA mAb and the anti-CD3 mAb, via the Duobody technique. It has been created by Pfizer.

The efficacy and safety of elranatamab has been investigated in MagnetisMM-3, a multicenter, open-label, single-arm, phase 2 study (NCT04649359). The study was performed on a cohort of 123 patients diagnosed with r/r MM unresponsive to at least one proteasome inhibitor, one immunomodulatory drug, and one anti-CD38 antibody, and must not have received prior BCMA-directed therapy. The primary endpoint was met with an ORR of 61.0%, including 35.0% with CR or better. With a median follow-up of 14.7 months, median duration of response, progression-free survival and OS (secondary endpoints) have not been reached. Fifteen-month rates were 71.5%, 50.9%, and 56.7%, respectively.

TEAEs were reported in all 123 patients treated with elranatamab, with grade 3 or 4 events reported in 70.7% of patients. TEAEs led to dose reductions and interruptions in 28.5% and 77.2% of patients, respectively. Hematological TEAEs, mostly neutropenia, were the most common (17.1%) TEAEs leading to dose reduction. The most frequent (≥20%) TEAEs leading to dose interruptions were infections (50.4%), most commonly coronavirus disease 2019 (COVID-19)-related (25.2%). Infections occurred in 69.9% of patients, including 6.5% of fatal infections. Of the 119 patients who received the two step-up priming-dose regimen, CRS occurred in 56.3% of patients. No CRS events grade 3 or higher were reported. ICANS occurred in 4 of 119 (3.4%) patients [109].

Elranatamab has been approved for the treatment of adult patients with r/r MM who have received at least four prior lines of therapy, including a proteasome inhibitor, an immunomodulatory agent, and an anti-CD38 monoclonal antibody. This indication is approved under accelerated approval based on response rate and durability of response. Pfizer continues to advance the MagnetisMM clinical program to expand Elrexfio into earlier lines of treatment, both as monotherapy and in combination with standard or novel therapies, as well as to compare the effectiveness of elranatamab and other types of therapies [110,111,112].

## 5. Challenges and Perspectives

The introduction of BsAb-based therapies has revolutionized the treatment of hematological malignancies. This method still has its limitations, including on target/off tumor killing, issues involving the choice of optimal affinities, and the toxicities associated with TCE-based treatments, which need to be solved to develop an effective and safe immunotherapy. However, the rapid development of various methods improving the properties of TCE, as well as combining TCE-based therapies with other forms of treatment, gives hope to find the cures for currently terminal diseases.

### 5.1. On Target/Off Tumor Killing

Choosing a proper TAA constitutes a fundamental issue regarding the safety of the agent. The presence of tumor-specific antigens enables differentiatioin of cancer cells from regular cells. However, the expression of an antigen specific only to the cancer cells is very rare. In most cases, TAA is present both on the tumor and normal cells. For example, CD19, often targeted in the treatment of hematological malignancies, is present on all B cells. Fortunately, the depletion of physiological B cells expressing TAAs is possible to manage [63].

Another important issue in developing a safe agent is the distinction between the levels of antigen expressed in cancer cells and normal cells. A prime example of this type of phenomenon is the BCMA, which is physiologically present on the surface of B cells but is largely overexpressed in tumor cells in MM [79].

### 5.2. Choice of Optimal Affinities

The affinity of a BiTE molecule to the target antigens constitutes an essential issue when developing a safe drug. One of the major factors increasing the BsAb’s potency is its affinity to TAA. This parameter can be improved, for example, by utilizing avidity in multivalent antibody formats like this used by Roche, which is, as previously mentioned, in a novel 2:1 format [18,99].

The strength of binding to the CD3 is especially important for the agent’s cytotoxicity. Generally, while affinity to TAA is typically higher and depends on tumor nature, the affinity of the fragment targeting CD3 should stay low, in order to avoid the T-cell activation in the absence of target cells, and to decrease the level of CRS associated with BsAb administration [63,113]. An alternative strategy to mitigate this syndrome is screening de novo for clones of anti-CD3 antibodies with sufficient cytotoxicity activity but depleted cytokine release properties. This can be managed, for example, by using a special platform that generates T-cell specific antibodies based on selected anti-CD3 arms with desired properties. However, further studies are required [114].

### 5.3. Tumor Immune Escape

Although the significant clinical activity of TCE-based therapies in hematology has been proven, still, a notable portion of patients do not respond to treatment, or they eventually relapse despite initial responses. Numerous means are contributing to the resistance to TCEs, but two major mechanisms can be distinguished, which include the involvement of immunosuppressive factors and the loss of the target antigen.

The importance of immune checkpoints in the tumor immune evasion has been proven in preclinical studies [115,116]. The application of the PD-1/PD-L1-blocking antibodies has shown a promising outcome in the treatment of various malignancies. Further research confirmed that the expression of inhibitory immune checkpoints, mainly PD-L1, was increased after TCE-based treatment. Both findings suggest that the combination of checkpoint inhibitors and TCE-based therapy is a favorable strategy to improve TCE efficacy [117,118,119].

Another factor contributing to the tumor immune escape is the activity of the immunosuppressive cells involved in the tumor microenvironment (TME). Tumors may utilize regulatory T-cells (Treg) (CD4/CD25/FOXP3) to create favorable conditions for their development by suppressing the tumor-specific immune response. Studies have shown that an increased Treg frequency negatively influences blinatumomab’s efficacy in the treatment of r/r B-ALL [113,120].

Myeloid-derived suppressor cells (MDSCs) have been identified as another cell population taking part in the TME [121]. For example, a subset of MDSCs, known as granulocytic MDSCs (CD13/CD16), largely contributes to the progression of MM [122]. Further research and a thorough understanding of immunosuppressive factors can help us develop new, improved regimens for targeted therapies, including not only TCE-based therapies, but also CAR-T therapies and more.

The loss of the targeted antigen is another major means of tumor immune evasion. It is a commonly observed scheme, that initially CD19-positive B-ALL patients achieve CR after blinatumomab treatment, but up to 50% of those patients experience CD19-negative relapse. Targeted antigen loss has also been identified as a factor negatively influencing the outcome of other T-cell-based cancer therapies, like CAR-T therapy [123,124].

Antigen loss can occur in a variety of mechanisms. Those include the loss of antigen expression and the loss of ability to recognize and bind to the said antigen, which can result from CD19 mutations, low CD19 RNA expression, and mutations in CD81 (forming signal transduction complex with CD19), were also identified as partial causes of loss of CD19 expression. Apart from decreased or suppressed CD19 expression, in both CD19-positive and CD19-negative relapses, it was observed that the presence of the CD19 RNA isoform ex2part was increased. The switch of RNA isoform results in shifting of the epitopes on the CD19 molecule, thus disrupting the binding of blinatumomab to CD19 [125].

Another interesting phenomenon is the lineage switch of tumor cells from B cells of the lymphoid lineage to cells from the myeloid lineage in B-ALL patients as a result of CD19-directed immunotherapy. In this case, CD19 expression is replaced with the upregulation of myeloid marker levels such as CD33. This event might be the consequence of the treatment-induced depletion of CD19+ tumor cells, which gives an advantage to subclones that do not express CD19 and gene rearrangements, such as lysine methyltransferase 2A/ALF transcription elongation factor 1 (KMT2A/AFF1) and zinc finger protein 384 (ZNF384) [126].

### 5.4. Immunotoxicity

The safety profile of TCE-based therapies is still a considerable concern when considering this treatment option. The results from the phase 3 TOWER study show that the most common AEs associated with TCE administration are neutropenia, infection, elevated liver enzyme, and neurotoxicity, namely, ICANS and CRS [87]. CRS and ICANS are the most dangerous in the group, being a serious, potentially lethal health hazard. Those two AEs are also proven to be the most common dose-limiting toxicities (DLTs) in TCE-based therapies [127,128].

CRS is a form of an uncontrolled systemic inflammatory response syndrome caused by robust activation of a large number of immune effector cells, which release high amounts of pro-inflammatory cytokines, primarily IL-6, which leads to further recruitment of more immune effector cells in a positive-feedback loop [129]. CRS occurs in association with CAR T cell therapy, bispecific TCEs, and monoclonal antibodies; however, it is most common in T-cell engaging therapies—CAR-T and TCEs. The clinical signs and symptoms of CRS involve multiple organ systems, with life-threatening complications, including fluid-refractory hypotension and cardiac dysfunction, respiratory failure, coagulopathy, and renal and liver failure. Fever is usually the first sign of CRS and develops before additional signs and symptoms. Low-grade CRS presents with a flu-like illness and patients often complain of fatigue, myalgia, and arthralgia [130]. The onset and severity of CRS is dependent on the immunologic agent and the degree of immune cell activation. For example, rituximab (anti-CD20) has been shown to induce CRS within hours of infusion, whereas CRS with T-cell therapies generally occurs within days to weeks after infusion [131,132,133]. It was found that strategic administration of blinatumomab during the course of therapy can be helpful in lowering the risk of CRS [124]. While mild CRS can be addressed by symptomatic treatment, more severe cases have to be treated with the use of additional agents. Dexamethasone is a steroid drug commonly administered before TCE-based therapy in order to prevent CRS [134]. However, the collected data regarding the steroid influence on TCE therapeutic effect is inconsistent. Brandl et al. did not report negative effects of dexamethasone on blinatumomab anti-tumor activity, while in a recent publication, Kauer et al. documented a profound inhibition of T-cell proliferation observed after the application of dexamethasone [135,136]. Apart from corticosteroids, there are other strategies available to mitigate the CRS in TCE-based therapies. It has been found that in blinatumomab treatment, the application of tocilizumab is useful for lowering the risk of CRS without inhibiting the therapeutic activity of blinatumomab [136].

ICANS is another unique treatment-related AE for patients treated with T-cell-engaging therapies. Pathogenic mechanisms responsible for neurotoxicity induced by T cell-based anti-cancer therapies are complex and incompletely understood. It is speculated that the activation of macrophages and monocytes causes the release of a great number of cytokines, which, in turn, increases vascular permeability and endothelial activation, leading to blood–brain barrier breakdown. There is evidence that the adhesion of T-cells to endothelial cells contributed to neurotoxicity induced by blinatumomab based on analyses of selected patients from five clinical trials and data from in vitro experiments [137]. Neurotoxicity occurs mainly in treatment cycle 1, and the risk of neurotoxicity is positively correlated with higher doses of blinatumomab. Symptoms of ICANS are variable and can appear ambiguous. Patients experience mild tremors, confusion, and agitation. A prominent and early feature of ICANS is speech and handwriting difficulties [138]. The most dangerous consequences of ICANS include seizures, fatal cerebral edema, and intracerebral hemorrhage [128]. Interventions used routinely to treat neurotoxicity include interrupting the treatment and the administration of dexamethasone [128]. It is suggested that agents inhibiting the adhesion between T cells and blood vessel endothelium could be effective in reducing the incidence of neurotoxicity [137].

### 5.5. New Agents with Improved Properties

Using the experience we gained with constructing BsAbs has consequently led to the creation of trispecific antibodies (TsAbs), which are able to bind three different antigens. Currently, there are no TsAbs evaluated in clinical trials, but preclinical studies have shown promising results for the application in hematological malignancies. Namely, Wu et al. have developed a TsAb with the premise of treating MM. The agent is a single antibody able to bind to CD38 on MM cells and CD3/CD28 on T-cells. The addition of a CD28 binding site to the classical TCE construct is meant to deliver a secondary signal to support the T-cell survival. The study has proven the agent effective against various myeloma cell lines in vitro, showing higher efficacy compared to the commercially available anti-CD38 agent-daratumumab [139].

Another strategy is to develop TsAbs targeting multiple TAAs. Multitargeted strategies may be effective, for example, in overcoming antigen loss, which could help cope with previously mentioned lineage switch issues. Developing a single drug that can simultaneously target multiple TAAs could also solve the problem of imprecise targeting of the cancer cells. TsAb molecules could be designed in order to target TAAs which occur in configurations typical for cancer cells, but not expressed on physiological cells. An example of this type of combination is a CD5+/CD23+ phenotype which occurs in many lymphoproliferative disorders, such as CLL and NHL [140].

### 5.6. Secreted TCE

Currently, TCE-based therapies are delivered passively, via intravenous or subcutaneous infusions, while native antibodies are actively secreted by physiological plasma cells in response to the presence of the targeted antigen. Implementing the TCE construct into the living cells of a patient’s organism to be secreted when and where necessary could help to maintain a stable therapeutic concentration sufficient for the anti-tumor effects, with no need for continuous infusion. Additionally, cells secreting TCEs could concentrate specifically in tumor regions, or even be the tumor cells themselves, thus providing targeted, tissue-specific drug administration, and as a result, reducing the systemic toxicity.

CAR-T cells and oncolytic viruses (OVs) are currently typed as the most promising way of alternative TCE delivery. Choi et al. proposed and developed the concept of CAR-T-BiTE, which was the CAR-T cells targeting EGFRvIII, which additionally secreted EGFR-targeting TCE antibodies [141].

BiTE-armed OVs are viruses targeting TAAs, equipped with a therapeutic transgene encoding TCEs. These modified viruses are designed to infect and replicate in tumor cells, ultimately inducing oncolysis, with additional T-cell activation, provided by TCE activity, leading to bystander killing of non-infected tumor cells [142].

### 5.7. Combination Approaches

#### 5.7.1. TCE in Combination with Chemotherapy

Hyperfractionated cyclophosphamide, vincristine, doxorubicin, and dexamethasone (Hyper-CVAD) is a commonly used chemotherapy regimen used for B-ALL therapy. It can serve as a basis for designing new treatment schemes. An ongoing phase 2 study (NCT02877303) is evaluating the regimen of Hyper-CVAD in combination with blinatumomab in patients with B-ALL. The treatment plan consists of four cycles of Hyper-CVAD followed by four cycles of blinatumomab and, so far, has promising results [143].

Two additional studies have also demonstrated the role of blinatumomab in combination with chemotherapy as front-line therapy in adults with newly diagnosed Ph- B-ALL with tolerable toxicity [144,145]. In addition, blinatumomab monotherapy after current standard-of-care chemotherapy—a regimen of rituximab combined with cyclophosphamide, doxorubicin, vincristine, and prednisolone (R-CHOP)—has achieved an ORR of 89% in patients with newly diagnosed high-risk DLBCL [146].

#### 5.7.2. TCE in Combination with Immune Checkpoint Inhibitors

As previously mentioned, the upregulation of immune checkpoints is one of the major mechanisms involved in resistance to BiTE therapy. The combination therapy with immune checkpoint inhibitors (ICI) can reactivate an exhausted immune response and improve anti-tumor activity; thus, several related clinical studies have been conducted. A small phase 1 study (NCT02879695) is evaluating the safety and tolerability of blinatumomab combined with nivolumab targeting PD-1 and/or ipilimumab targeting cytotoxic T-lymphocyte antigen 4 (CTLA-4) in patients with r/r B-ALL. Eight patients were enrolled, with a median bone marrow blast percentage of 73%. Preliminary results have shown that the CR rate was 80%, while all of the patients were MRD-negative [147]. Another ongoing phase 1–2 study (NCT03160079) has also shown blinatumomab in combination with a PD-1 inhibitor, pembrolizumab, to be effective and relatively safe in patients with r/r B-ALL. Encouraging anti-leukemic activity has been seen in the majority of treated patients—CR rate of 79% (11/14)—and most CRs are MRD negative (71%), supporting the application of combination therapies involving these two types of agents [148].

#### 5.7.3. TCE in Combination with CAR-T Therapy

CAR-T cell therapy has revolutionized cancer treatment. CAR-T cell therapy uses T-cells genetically modified with the ability to express CAR—a synthetic receptor engineered with the premise to precisely recognize a specific antigen and activate cytotoxic response of the T-cells. While CAR-T cell therapies and TCE-based therapies are direct competitors in performance in treating blood cancers, there is evidence that those two treatment options can join their forces in a combination approach.

In an encouraging outcome published by Shalabi et al., prescribing blinatumomab alongside anti-CD22, CAR-T cell therapy led to the complete eradication of tumor cells and the prolonged life span of patients who relapsed after anti-CD22 CAR-T cell monotherapy. Interestingly, blinatumomab induced CAR-T cell expansion, proliferation, persistence, and cytokine production, which led to the potent action against tumor cells [149].

#### 5.7.4. TCE in Combination with Oncolytic Viruses (OVs)

OVs are viruses modified to target tumor cells and induce their lysis, which, in turn, promotes local T-cell response, making this type of therapy convenient to combine with TCEs [150]. Additionally, as OVs are designed to be selective towards particular tissues, implementing the TCE construct into OV could also reduce drug toxicity [151]. However, due to the nature of OVs, while this treatment regimen could be effective against localized tumors such as lymphomas, it is not suitable for the majority of hematological malignancies.

In an in vitro study, Scott et al. have shown that OVs with implemented CD3 × EpCAM TCE construct effectively promote T-cell anti-tumor activity, without an antagonistic relationship between the T-cell population and virulence, which was a suspected issue [150].

Speck et al. tested the OV-TCE in vivo in mouse tumor models. Collected data revealed that an OV equipped with CD3 × CD20 TCE was more effective against tumor cells than a control OV armed with an irrelevant TCE (CD3 × carcinoembryonic antigen (CEA)) or TCE alone. Interestingly, it was also found that the activity of CD3 × CD20-OV induced PD1 upregulation, which suggest further study regarding the addition of a suitable ICI in this therapy regimen [152].

## 6. Conclusions

Despite advances in the field of immunotherapy, the needs of many cancer patients remain unmet. However, the strategy of directing the cellular immune response onto cancer cells—in particular, the T-cell-redirecting therapeutics, including CAR-T cell and TCE-based therapies—has shown its potential in the treatment of hematological malignancies. The successes achieved with blinatumomab and other approved TCEs, as well as the results of ongoing clinical studies, prove TCE-based therapies as an optimal therapeutic approach. However, as much as this type of therapy has its benefits, it also has numerous shortcomings. A significant fraction of patients do not benefit from TCE-based immunotherapy in the long term, due to mechanisms of tumor immune evasion, such as the loss of the targeted antigen, or T-cell anergy and exhaustion caused by upregulation of immune checkpoints. What is more, the TEAEs, such as CRS or ICANS, although less probable to occur compared to CAR-T cell-based therapies, remain a considerable issue when regarding this treatment option. Combinatory approaches of TCE-based therapies with other therapeutic agents, such as standard chemotherapy or ICIs, are currently being researched in order to overcome those issues. Moreover, improving the molecular structure of TCEs might enhance their affinity, flexibility, and half-life. Although the deeper understanding of the TCE technology is required for the establishment of a safe and effective treatment option, currently collected data give hope to find the cures for currently terminal hematological malignancies.

## Figures and Tables

**Figure 1 cancers-16-01580-f001:**
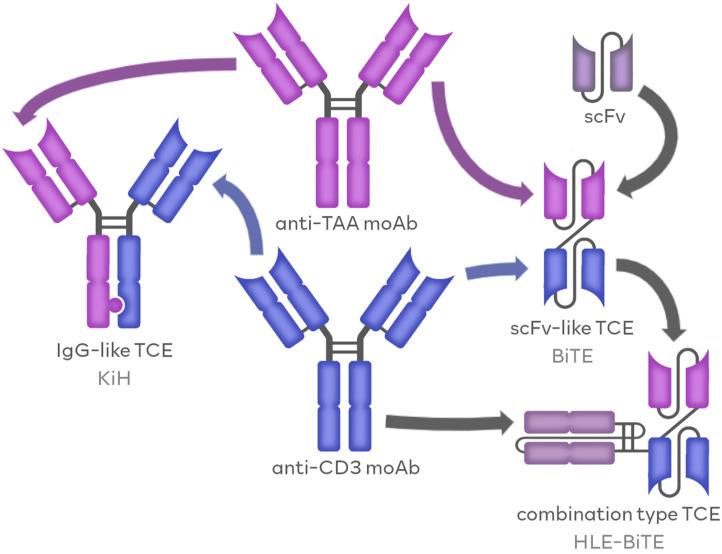
The molecular structure of TCE molecules. TCE molecules combine the features of anti-CD3 moAbs and anti-TAA moAbs. TCEs can resemble naturally occurring antibodies, in IgG-like formats (e.g., KiH), or in Fv-like formats (e.g., BiTE), and they can be composed of synthetic scFvs. Combining the features of IgG-like and Fv-like formats resulted in the development of combination-type TCEs, like half-life extended BiTE (HLE-BiTE).

**Figure 2 cancers-16-01580-f002:**
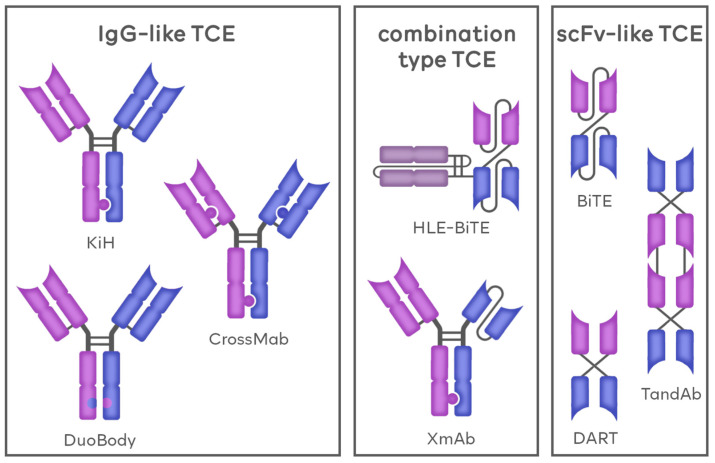
Types of TCEs according to their molecular structure. TCEs can be divided into three groups according to their molecular structure. Immunoglobulin G-like (IgG-like) TCEs share similarities with antibodies naturally occurring in the human body. The most popular methods of generating the IgG-like TCEs include knob-in-a-hole (KiH), CrossMab and DuoBody. The single-chain variable fragment-like (scFv-like) TCEs agents are constructed on the basis of synthetic scFv molecules and include for example BiTE, DART and TandAb platforms. Combination type TCEs share similarities.

**Figure 3 cancers-16-01580-f003:**
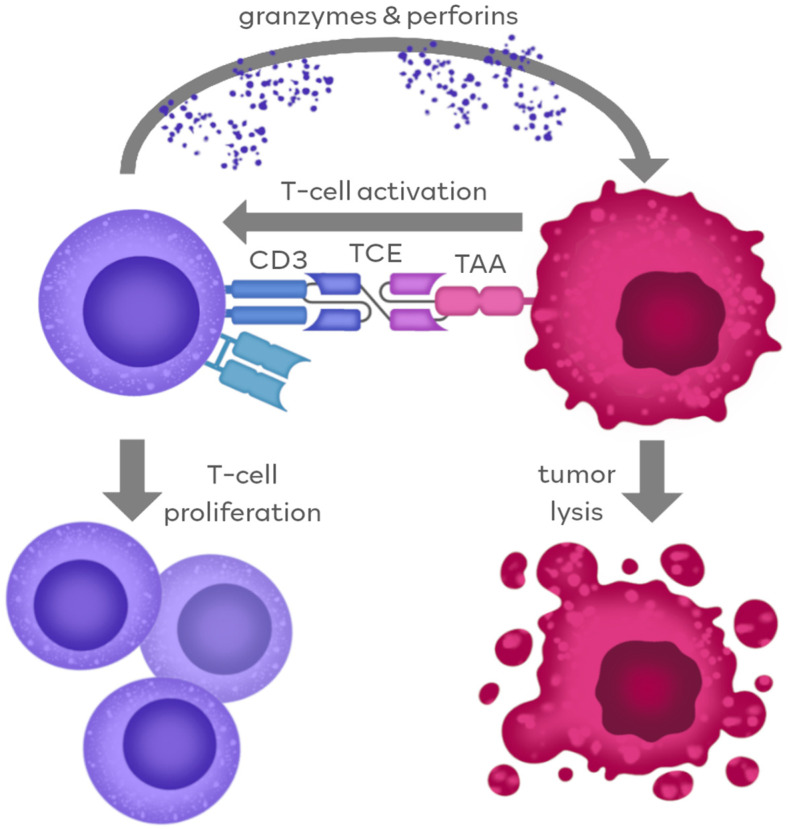
TCE action through antigen binding. TCE molecule simultaneously binds to both TAAs and CD3, allowing for the formation of the cytolytic synapse. Association with the CD3 triggers the induction of an activating signal for the T-cell cytotoxic action, including the release of granzymes and perforins, ultimately resulting in tumor cell lysis. Additionally, the stimulation by TCE triggers the proliferation of the T-cell compartment.

**Table 1 cancers-16-01580-t001:** All currently approved TCEs in hematology (March 2024).

Drug Name	Targets	Indications	Platform	First Approved Date (Country)	Confirmatory Study	Primary Endpoint	The Most Common AEs
Blinatumomab	CD3/CD19	r/r B-ALL	BiTE	December 2014 (USA)	NCT01207388	CRR: 78%	Pyrexia: 89% Neurological events: 53% Headache: 38%
Mosunetuzumab	CD3/CD20	r/r FL	KiH	June 2022 (EU)	NCT02500407	CRR: 60%	Neutropenia: 28% CRS: 27% Hypophosphatemia: 23%
Teclistamab	CD3/BCMA	r/r MM	DuoBody	August 2022 (EU)	NCT04557098	ORR: 63%	CRS: 72% Neutropenia: 71% Anemia: 51%
Glofitamab	CD3/CD20	DLBCL	2:1 Crossmab	March 2023 (Canada)	NCT03075696	CRR: 39%	CRS: 66% Neutropenia: 38% Anemia: 31%
Epcoritamab	CD3/CD20	DLBCL	DuoBody	May 2023 (USA)	NCT03625037	ORR: 63%	CRS: 50% Pyrexia: 24% Fatigue: 23%
Talquetamab	CD3/GPRC5D	r/r MM	DuoBody	August 2023 (USA)	NCT03399799	ORR: 70% *, 64% **	CRS: 77% *, 80% ** Skin-related events: 67% *, 70% ** Dysgeusia: 63% *, 57% **
Elranatamab	CD3/BCMA	r/r MM	DuoBody	August 2023 (USA)	NCT04649359	ORR: 61%	CRS: 58% Anemia: 49% Neutropenia: 49%

AEs—adverse events, r/r B-ALL—relapsed or refractory precursor B-cell acute lymphoblastic leukemia, r/r FL—relapsed or refractory follicular lymphoma, DLBCL—diffuse large B-cell lymphoma, r/r MM—relapsed/refractory multiple myeloma, BiTE—bispecific T-cell engager, KiH—knob-in-a-hole, CRR—complete response rate, ORR—overall response rate, CRS—cytokine release syndrome. *—subcutaneous talquetamab, 405 μg weekly; **—subcutaneous talquetamab, 800 μg every 2 weeks.

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
