# Peer review of "T-Cell Engagers—The Structure and Functional Principle and Application in Hematological Malignancies"

_cancers, 2024, doi:10.3390/cancers16081580_

Round 1
Reviewer 1 Report
Comments and Suggestions for Authors
This review article is excellent and highly informative for readers. However, manuscript preparation contains some issues to be corrected including abbreviations and figure legends.
Minor comments:
1. In this article, although there are many abbreviations properly used, still many abbreviations to be defined; for example, LAG-3 (line 87), HLE (line 109 and Figure 1), r/r (line202), TAEs (line 216), XmAb (line 230), Xencor (line 231), DLBCL and FL (line 236), EpCAM (line 3 01), HER2 (line 320), FLT3 (line 328), PI3K (line 499), GPRCD5 (line 580), Treg (line 689), KMT2A/AFF1 (line 717), ZNF 384 (line 718), and CEA (line 860).
2. Is the abbreviation SAEs (line 45) needed?
3. Conversely, the full term writings such as “immunoglobulin G” (line 112), “variable fragment” (line 115), “constant region of the heavy chain” (line 168), Half life extended BiTE” (line 223), “diffuse large B-cell lymphoma” and “follicular lymphoma (line 346), “The FMS-like tyrosine kinase” (line 392) are not needed.
4. Figure legends for each Figure appear to be a part of the text. Please correct these legends in smaller letter size, including proper abbreviation.
5. Blinatumomab etc, should be written as blinatumomab except for the beginning of the sentence in all cases. Cytotoxic (line 86) should be as cytotoxic. Similarly, Open-label (line 400) should be as open-label.
6. in vitro (line 779)→in vitro
7. Is BsaAbs (line 82) a typo?
8. R/R should be as r/r in all cases.
9. >2 (line 375)→≧2
10. What is (siglec) (line 381)?
11. Is “for injection” (line 419) needed?
12. Aug. and Sept should be as August and September in all cases.
13. Are MDRs (line 448) typos of MRD?
Reviewer 2 Report
Comments and Suggestions for Authors
The article entitled “T-Cell Engages – The Structure and Functional Principle and Application in Hematological Malignancies” explains all the knowledge about the immunotherapy in the setting of hematological malignancies such as Acute Lymphoblastic Leukemia, Diffuse Large B-Cell Lymphoma, LBCL, Follicolar Lymphoma, Multiple Myeloma. This review is very well-written. The topic is vast and complex but the authors have structured it very well driving from construction of novel bispecific antibodies (T-cell engagers) to clinical applications. In the clinical setting there are all concepts about the action mechanism in association with the adverse effect and their treatments. I think that this editorial work is well made in terms of references. Therefore, I think that this article is suitable for publication in its current version.
